# Design and Analysis of a Hybrid Displacement Amplifier Supporting a High-Performance Piezo Jet Dispenser

**DOI:** 10.3390/mi14020322

**Published:** 2023-01-27

**Authors:** Shuai Zhou, Peng Yan

**Affiliations:** Key Laboratory of High-Efficiency and Clean Mechanical Manufacture, Ministry of Education, School of Mechanical Engineering, Shandong University, Jinan 250061, China

**Keywords:** compliant mechanism, PZT actuator, high performance, piezo jet dispenser

## Abstract

In this study, a compliant amplifier powered by a piezoelectric stack is designed to meet high-performance dispensing operation requirements. By studying the issue of low frequency bandwidth on the traditional bridge-type amplifier mechanism, we propose a displacement amplifier mechanism, hybrid bridge-lever-bridge (HBLB), that enhances its dynamic performance by combining the traditional bridge-type and lever mechanism. A guiding beam is added to further improve its output stiffness with a guaranteed large amplification ratio. An analytical model has been developed to describe the full elastic deformation behavior of the HBLB mechanism that considers the lateral displacement loss of the input end, followed by a verification through a finite element analysis (FEA). Results revealed that the working principle of the HBLB optimizes the structural parameters using the finite element method. Finally, a prototype of the displacement amplifier was fabricated for performance tests. Static and dynamic test results revealed that the proposed mechanism can reach a travel range of 223.2 μm, and the frequency bandwidth is 1.184 kHz, which meets the requirements of a high-performance piezo jet dispenser.

## 1. Introduction

Piezoelectric materials play an important role in the field of flexible electronics [1,2,3,4], precision engineering [5,6,7], and microelectronics packaging [8,9,10,11]. Piezoelectric materials can deform under the action of electrical flow, which is called piezoelectric effect [12]. The piezoelectric stacks made by connecting several piezoelectric ceramic pieces in series can efficiently converts electrical energy into mechanical energy. Piezo jet dispensers powered by piezoelectric stacks have become popular in the microelectronics packaging industry because of their high frequency, large output force, and rapid response of piezoelectric stacks [13,14,15].

However, the output displacement of piezoelectric stacks is generally small [16]. To meet the required travel range for application in jet dispensers [17], the output stroke of the piezoelectric stacks must be enlarged by the displacement amplification mechanism. The compliant displacement amplifier mechanism has attracted attention in the design of piezoelectric actuators owing to its low friction, high resolution, and high precision [18,19,20,21], compared to traditional amplifier mechanisms. Various compliant amplifier mechanisms, such as the lever-type [22,23,24], bridge-type [20,25], and Scott-Russell [26] mechanisms, have been widely applied. Lu et al. presents a diamond-type micro-displacement amplifier with a natural frequency of 342 Hz [27]. The micro-displacement amplifier was applied in jet dispenser [28]. Bu et al. presents a new jetting dispenser consisting of a displacement amplifying mechanism with a corner-filleted flexure hinge. The natural frequency of the displacement amplifying mechanism is 469 Hz [29]. Because a single-stage amplifier mechanism usually confronts problems, such as large self-size, small amplification ratio, and low natural frequency, extensive research focusing on multi-stage amplifier mechanism has been conducted in recent years. Wang et al. presented an arch-shaped bridge-type displacement amplifier with a higher resonant frequency by enhancing the lateral stiffness [30]. Zhu et al. proposed an innovative hybrid displacement amplifier by combining two Scott-Russell mechanisms with a half-bridge mechanism, which had better dynamic characteristics [19]. Wu et al. proposed a new bridge-type displacement amplifier. The resonant frequency was improved by embedding a Scott-Russell mechanism [31].

These displacement amplifier mechanisms often have a proper amplification ratio capable of large working strokes; however, their low resonant frequency limits their application in high-performance jet dispensers. Therefore, a hybrid amplifier mechanism with multi-stage amplification is proposed. This hybrid amplifier mechanism replaces the input end of the traditional bridge-type (BT) mechanism with a lever mechanism to enhance the weak dynamic bandwidth of the bridge-type (BT) mechanism while retaining a large amplification ratio. The resonant frequency of the hybrid displacement amplifier mechanism has been significantly increased. Additionally, a guiding beam mechanism is introduced into the output ends of the hybrid amplifier mechanism, which is applied to restrain the motion of the output end and enhance the stiffness and harmonic mode along the working direction.

The rest of this paper is organized as follows: the mechanical design of the hybrid amplifier mechanism is discussed in Section 2; the corresponding analytical model is given in Section 3; finite element analysis (FEA) optimization and verification are presented in Section 4; the experimental verification is presented in Section 5; and, finally, the conclusions are presented in Section 6.

## 2. Mechanical Design

### 2.1. Design Motivation

Owing to the large output force and fast response of piezoelectric actuators, piezo jet dispensers have been widely applied in the microelectronics packaging industry to jet high-viscosity glue with a high working efficiency. The jet dispenser mainly consists of a nozzle, needle, fluid chamber, and oil sealing, as shown in Figure 1, where the needle is threaded to the output of the piezoelectric actuator. During the jetting process, the needle driven by a piezoelectric actuator pushes part of the fluid in the fluid chamber jetting through the nozzle. To realize efficient jetting, it is necessary to design a piezoelectric actuator with the desired work stroke [17] and high working bandwidth, which requires a compliant amplifier mechanism with a proper amplification ratio and high resonant frequency.

In contrast, a displacement amplifier based on a compliant mechanism encounters a contradiction between the amplification ratio and the resonant frequency. As an extremely common displacement amplifier, the bridge-type amplifier mechanism (BTAM), as shown in Figure 2a, has the advantages of a large amplification ratio and small size; however its dynamic performance is poor because of its low lateral stiffness [32]. Based on this, Ling [33] proposed an improved bridge-type compliant mechanism that avoids inertial motion of the BTAM by fixing four guiding flexible beams to the input end, as shown in Figure 2b. This was applied to develop a piezoelectric flow control valve with a rapid response and large flow rate because the improved bridge-type compliant mechanism has a high resonant frequency.

### 2.2. Mechanical Structure

Inspired by the improved bridge-type compliant mechanism [33], a novel compliant amplifier mechanism combining the lever mechanism with the input end of the BTAM, hybrid bridge-lever-bridge (HBLB) type, is proposed, as depicted in Figure 3a. The lever mechanism limits the lateral motion of the BTAM, thereby making the HBLB mechanism a high-natural frequency mechanism. Moreover, the input port is closer to the lower-bridge mechanism in this design, which ensures the required amplification ratio, as shown in Figure 3b. Further, a guiding mechanism is also applied in it to further improve the output stiffness. The working principle of the HBLB mechanism is further discussed in Section 4.

One side of the piezoelectric stacks of the traditional lever mechanism is fixed to the ground, and the other side is in contact with the arm of the lever mechanism, which causes an unintended lateral force to be imposed on the stacks. In the HBLB mechanism, both sides of the piezoelectric stacks are in direct contact with the input end of the designed mechanism, of which the contact force between the input of the amplifier and the output of the stacks is along the direction of the piezoelectric stacks under the assumption of no friction. Thus, the piezoelectric stacks are well protected, as shown in Figure 4.

## 3. Theoretical Analysis

In this section, the compliant matrix method is adopted to simplify the modeling procedure of the HBLB mechanism. We assumed that the small linear deformation appears only in flexure hinges, and others are regarded as rigid structures without deformation.

In this study, a corner-filleted flexure hinge was applied to maintain a compact size, and we assumed that hinge deformation only occurs in the plane because the amplification mechanism is planar and sufficiently thick [34]. The deformation $\delta X_{i}$, $\delta Y_{i}$, $\delta \theta_{i}$ of the corner-filleted flexure hinge *i* (*i* = A, B, ⋯, Q) under forces $F_{X}$, $F_{Y}$, $M_{Z}$ can be obtained as:(1)$$\begin{array}{c} \left[\begin{array}{c} \delta X_{i}\\ \delta Y_{i}\\ \delta \theta_{i} \end{array}\right]=\left[\begin{array}{ccc} C_{{}_{11}}^{i} & 0 & 0\\ 0 & C_{{}_{22}}^{i} & C_{{}_{23}}^{i}\\ 0 & C_{{}_{32}}^{i} & C_{{}_{33}}^{i} \end{array}\right]\left[\begin{array}{c} F_{xi}\\ F_{yi}\\ M_{i} \end{array}\right], \end{array}$$
where, the compliance factors in the matrix can be referred to in [35].

Owing to the symmetric structure of the HBLB mechanism, we establish only half of the structure, and the mechanical analysis is illustrated in Figure 5. $l_{i},b_{i},t_{i}$ (*i* = b,c,h) denote the length, width, and thickness of the flexure hinge, respectively. $l_{i}$ (*i* = 1, 2, ⋯, 9) denote the relative distance between the hinges. Considering the simplification of the calculation, the bridge arm, guiding beam, and lever mechanism-based input end were analyzed before obtaining the overall model of the HBLB mechanism in this analysis. Furthermore, considering the deformation of the input end, an analytical model based on the beam constraint model was established. Subsequently, FEA and experiments were conducted to verify the proposed modeling method.

### 3.1. Bridge-Arm

A force analysis of the lower-bridge mechanism is shown in Figure 6a. Based on the force balance conditions of link AB, the relationships can be obtained as follows:(2)$$\begin{array}{c} \left\{\begin{array}{c} F_{Ax}=F_{Bx}\\ F_{Ay}=F_{By}\\ M_{A}=F_{Bx}l_{1}-F_{By}l_{2}-M_{B}, \end{array}\right. \end{array}$$
where $F_{Bx},F_{By}$, $M_{B}$ and $F_{Ax},F_{Ay}$, and $M_{A}$ are the forces and moments exerted on the linkage by hinge B and A, respectively. Based on the compliant matrix method, the output displacements of the lower-bridge mechanism are the result of the deformation of the flexure hinges and rotational motion of the rigid linkage. Therefore, we can obtain the relative displacement of link AB as:(3)$$\left\{\begin{array}{c} \Delta X_{AB}=\delta X_{A}+l_{1}\delta \theta_{A}+\delta X_{B}\\ \Delta Y_{AB}=\delta Y_{A}+l_{2}\delta \theta_{A}+\delta Y_{B}\\ \Delta \theta_{AB}=-\delta \theta_{A}+\delta \theta_{B}, \end{array}\right.$$
where, $\delta X_{i},\delta Y_{i},\delta \theta_{i}$ (*i* = A,B) are scalar quantities whose values can be obtained using Equation (Equation 1).

Similar to the lower-bridge mechanism, the force analysis of the upper-bridge mechanism is illustrated in Figure 6b. According to the principle of the force equilibrium theory, we can obtain the following equations:(4)$$\left\{\begin{array}{c} F_{Hx}=F_{Jx}\\ F_{Hy}=F_{Jy}\\ M_{J}=F_{Hx}l_{7}-F_{Hy}l_{8}-M_{H}. \end{array}\right.$$

The relative displacement of link HJ also can be represented as:(5)$$\begin{array}{c} \left\{\begin{array}{c} \Delta X_{HJ}=\delta X_{H}+l_{7}\delta \theta_{J}+\delta X_{J}\\ \Delta Y_{HJ}=\delta Y_{H}+l_{8}\delta \theta_{J}+\delta Y_{J}\\ \Delta \theta_{HJ}=\delta \theta_{H}-\delta \theta_{J}. \end{array}\right. \end{array}$$

### 3.2. Guiding Beam

The force analysis of the guiding beam mechanism is shown in Figure 7. The guiding mechanism is modeled based on the Euler-Bernoulli beam theory; the bending equation of a flexible beam can be written as follows:(6)$$\begin{array}{c} w=\;\int \int \;\left(\frac{M\left(x\right)}{EI}dx\right)dx+mx+n\\ \theta =\int \frac{M\left(x\right)}{EI}dx+m, \end{array}$$
where *w* is the perpendicular distance to the neutral axis, *I* is the moment of inertia, and *E* is the Young’s modulus. $M\left(x\right)$ is the moment applied at position x about the neutral axis. Because the lateral displacement is significantly smaller than the vertical displacement of the guiding mechanism, the effect of $F_{Px}$ can be ignored, and $M\left(x\right)$ can be simplified as:(7)$$M\left(x\right)=F_{Py}x-M_{P},$$
where $F_{Py}$ and $M_{P}$ represent the force and moment applied at point P, respectively. According to the boundary constraints, both the deflection and rotation angles of the fixed end Q are equal to zero, and the rotation angle of point P is also equal to zero. Therefore, $w_{P}$ can be derived by solving the Equation (Equation 6) as:(8)$$w_{P}=\frac{F_{Py}\times {l_{9}}^{3}}{24EI},$$
where, $F_{Py}$ = $F_{Jy}$ according to the force equilibrium theory.

### 3.3. Lever Mechanism-Based Input End

The force analysis of the lever mechanism-based input end is shown in Figure 8. $F_{B^{\prime }x},F_{B^{\prime }y},\mathrm{and}\;M_{B^{\prime }}$ are the interaction forces applied from point B in the lower-bridge mechanism. Similarly, $F_{H^{\prime }x},F_{H^{\prime }y},\mathrm{and}\;M_{H^{\prime }}$ are the interaction forces applied from point H in the upper-bridge mechanism. Based on the force equilibrium conditions of lever mechanism C − B′ − D − H′, the following relationships can be obtained as:(9)$$\left\{\begin{array}{c} F_{Cx}=F_{H^{\prime }y}-F_{B^{\prime }y}\\ F_{Cy}=F_{in}-F_{H^{\prime }x}-F_{B^{\prime }x}\\ M_{C}=F_{in}\times l_{5}+F_{B^{\prime }x}\times l_{6}-F_{H^{\prime }x}\times l_{4}-F_{B^{\prime }y}\times l_{3}+F_{H^{\prime }y}\times l_{3}-M_{B^{\prime }}-M_{H^{\prime }} \end{array}\right.$$
where $F_{in}$ denotes the driving force produced by the piezoelectric stacks. Similar to the analysis above, the deformation of points B′ and H′ in the lever mechanism can be derived as:(10)$$\begin{array}{c} \left\{\begin{array}{c} \Delta X_{CB^{\prime }}=l_{4}\delta \theta_{C}+\delta Y_{C}\\ \Delta X_{CH^{\prime }}=l_{6}\delta \theta_{C}+\delta Y_{C}\\ \Delta Y_{CB^{\prime }}=\Delta Y_{CH^{\prime }}=l_{3}\delta \theta_{C}-\delta X_{C}\\ \Delta \theta_{CB^{\prime }}=\Delta \theta_{CH^{\prime }}=\delta \theta_{C}. \end{array}\right. \end{array}$$

Referring to Equation (Equation 6), the angle and vertical deformation of points B′, D, and H′ in the lever mechanism-based input end can be calculated as:(11)$$\begin{array}{cc} \theta_{B^{\prime }} & =\frac{-F_{Cy}{l_{4}}^{2}/2-M_{C}l_{4}}{EI}\\ w_{B^{\prime }} & =\frac{-F_{Cy}{l_{4}}^{3}/6-M_{C}l_{4}^{2}/2}{EI}\\ \theta_{D} & =\theta_{B^{\prime }}+\frac{-\left(F_{Cy}+F_{B^{\prime }x}\right)\left({l_{5}}^{2}-{l_{4}}^{2}\right)/2-\left(M_{C}+M_{B^{\prime }}-F_{B^{\prime }x}l_{4}\right)\left(l_{5}-l_{4}\right)}{EI}\\ w_{D} & =w_{B^{\prime }}+\theta_{B^{\prime }}\left(l_{5}-l_{4}\right)\\ \; & -\frac{\left(F_{Cy}+F_{B^{\prime }x}\right)\left({l_{5}}^{3}-{l_{4}}^{3}\right)/6+\left(M_{C}+M_{B^{\prime }}-F_{B^{\prime }x}l_{4}\right)\left({l_{5}}^{2}-{l_{4}}^{2}\right)/2}{EI}\\ \theta_{H^{\prime }} & =\theta_{D}+\frac{F_{H^{\prime }x}\left({l_{6}}^{2}-{l_{5}}^{2}\right)/2-\left(M_{H^{\prime }}-F_{H^{\prime }x}l_{6}\right)\left(l_{6}-l_{5}\right)}{EI}\\ w_{H^{\prime }} & =w_{D}+\theta_{D}\left(l_{6}-l_{5}\right)\\ \; & -\frac{F_{H^{\prime }x}\left({l_{6}}^{3}-{l_{5}}^{3}\right)/6+\left(M_{H^{\prime }}-F_{H^{\prime }x}l_{6}\right)\left({l_{6}}^{2}-{l_{5}}^{2}\right)/2}{EI}. \end{array}$$

### 3.4. Overall Mechanism

Based on the deformation compatibility condition, the motion deformation relationship can be summarized as follows:(12)$$\left\{\begin{array}{c} \delta X_{A}+l_{1}\delta \theta_{A}+\delta X_{B}=l_{4}\delta \theta_{C}+\delta Y_{C}-w_{B^{\prime }}\\ \delta Y_{A}+l_{2}\delta \theta_{A}+\delta Y_{B}=l_{3}\delta \theta_{C}-\delta X_{C}\\ -\delta \theta_{A}+\delta \theta_{B}=\delta \theta_{C}-\theta_{B^{\prime }}\\ \delta X_{H}+l_{1}\delta \theta_{J}+\delta X_{J}=l_{6}\delta \theta_{C}+\delta Y_{C}-w_{H^{\prime }}\\ \delta Y_{H}+l_{2}\delta \theta_{J}+\delta Y_{J}=l_{3}\delta \theta_{C}-\delta X_{C}+w_{p}\\ \delta \theta_{H}-\delta \theta_{J}=\delta \theta_{C}-\theta_{H^{\prime }}. \end{array}\right.$$

Therefore, the value of $F_{ix},F_{iy},M_{i}$ (*i* = B′, C, H′) can be obtained by passing Equation (Equation 12). Subsequently, the lateral deformation loss of the input end can be approximately obtained based on the beam constraint model. The lateral deformation of the input end can be considered as the addition of the normalized lateral compression caused by axial force $\delta_{xe}$ and normalized lateral displacement caused by beam bending $\delta_{xk}$. The normalized lateral displacement is derived as follows [36]:(13)$$\begin{array}{cc} \delta_{x} & =\delta_{xe}+\delta_{xk}\\ \; & \approx \frac{t^{2}f}{12}+\frac{\left(420-f\right)}{700}\delta_{i}^{2}-\frac{\left(70-f\right)}{700}\delta_{i}\theta_{i}+\frac{\left(420-f\right)}{700}{\theta_{i}}^{2}, \end{array}$$
where $t=\frac{T}{L}$ is the normalized thickness of the rigid body, $f=\frac{FL^{2}}{EI}$ is the normalized lateral force applied on the input end, $\delta_{i}=\frac{w_{i}}{L}$ (*i* = B′, D, H′) is the normalized vertical deformation of the input end, and $\theta_{i}$ is the angle of the input end. The lateral deformation loss $\Delta \delta_{i}=L\delta_{xi}$ of points B′, D, and H′ in the lever mechanism-based input end can be obtained as Equation (Equation 14):(14)$$\begin{array}{c} \begin{array}{c} \Delta \delta_{B^{\prime }}=\frac{l_{4}}{EI}\left(\frac{T^{2}F_{Cx}}{12}+\frac{\left(420EI-F_{Cx}{l_{4}}^{2}\right)}{700}\frac{w_{B^{\prime }}^{2}}{{l_{4}}^{2}}-\frac{\left(70EI-F_{Cx}{l_{4}}^{2}\right)}{700}\frac{w_{B^{\prime }}\theta_{B^{\prime }}}{l_{4}}+\frac{\left(420EI-F_{Cx}{l_{4}}^{2}\right)}{700}\theta_{B^{\prime }}^{2}\right)\\ \Delta \delta_{D}=\frac{\left(l_{5}-l_{4}\right)}{EI}\left(\frac{T^{2}F_{Ey}}{12}+\frac{\left(420EI-F_{Ey}{\left(l_{5}-l_{4}\right)}^{2}\right)}{700}\frac{{\left(w_{D}-w_{B^{\prime }}\right)}^{2}}{{\left(l_{5}-l_{4}\right)}^{2}}-\frac{\left(70EI-F_{Ey}{\left(l_{5}-l_{4}\right)}^{2}\right)}{700}\frac{\left(w_{D}-w_{B^{\prime }}\right)\theta_{D}}{\left(l_{5}-l_{4}\right)}+\frac{\left(420EI-F_{Ey}{\left(l_{5}-l_{4}\right)}^{2}\right)}{700}\theta_{D}^{2}\right)\\ \Delta \delta_{H^{\prime }}=\frac{\left(l_{6}-l_{5}\right)}{EI}\left(\frac{T^{2}F_{H^{\prime }y}}{12}+\frac{\left(420EI-F_{H^{\prime }y}{\left(l_{6}-l_{5}\right)}^{2}\right)}{700}\frac{{\left(w_{H^{\prime }}-w_{D}\right)}^{2}}{{\left(l_{6}-l_{5}\right)}^{2}}-\frac{\left(70EI-F_{H^{\prime }y}{\left(l_{6}-l_{5}\right)}^{2}\right)}{700}\frac{\left(w_{H^{\prime }}-w_{D}\right)\theta_{H^{\prime }}}{\left(l_{6}-l_{5}\right)}+\frac{\left(420EI-F_{H^{\prime }y}{\left(l_{6}-l_{5}\right)}^{2}\right)}{700}\theta_{H^{\prime }}^{2}\right). \end{array} \end{array}$$

Furthermore, the values of $Y_{out}$ and $X_{in}$ can be expressed as:(15)$$\begin{array}{c} Y_{out}=w_{P}-\Delta \delta_{B^{\prime }}-\Delta \delta_{D}-\Delta \delta_{H^{\prime }}\\ X_{in}=\Delta X_{CD}-w_{D}. \end{array}$$

The equivalent stiffness and the displacement amplification ratio of the designed mechanism can be derived as:(16)$$\begin{array}{c} K=\frac{F_{in}}{Y_{out}}=\frac{24EIF_{in}}{F_{H^{\prime }y}{l_{9}}^{3}-24EI\left(\Delta \delta_{B^{\prime }}+\Delta \delta_{D}+\Delta \delta_{H^{\prime }}\right)}\\ \kappa =\frac{Y_{out}}{2X_{in}}=\frac{F_{H^{\prime }y}{l_{9}}^{3}}{48EI\left(\left(C_{32}^{C}F_{Cy}+C_{33}^{C}M_{C}\right)l_{5}+\left(C_{22}^{C}F_{Cy}+C_{23}^{C}M_{C}\right)-w_{D}\right)}. \end{array}$$

## 4. Finite Element Analysis

In this section, finite element simulation is implemented using the commercial software ANSYS to analyze and optimize the structural parameters of the HBLB mechanism and verify the established mathematical model above.

### 4.1. Dimensional Parameter Analysis and Optimization

Recalling the mathematical modeling, we select four key structural parameters of the HBLB mechanism to analyze their influence on the output displacement and natural frequency, as shown in Figure 5. These structural parameters are the hinge thickness and positions of the lower-bridge and lever mechanisms, of which the optimized ranges are presented in Table 1. The sign of the value means the direction relative to the middle position. The material used for this hybrid mechanism was aluminum alloy Al7075, and the physical and mechanical parameters of this alloy are listed in Table 2. The technical parameters of the piezoelectric stacks are listed in Table 3. In the simulation, the fixed constraint was assigned to six through-holes, and the input forces of 400 N were applied to the input surface. The input and output displacements as well as the natural frequencies were obtained.

Figure 9 shows the effects of $t_{b}$ and $l_{1}$ on the output displacement and natural frequency, obtained through a simulation, when $t_{c}$ is set to 1 mm and $l_{3}$ is 2 mm. The maximum output displacement occurs when $t_{b}$ is approximately 1.1 mm. The natural frequency increases with an increase in $t_{b}$ or a decrease in $l_{1}$.

Figure 10 shows the effects of $t_{c}$ and $l_{3}$ on the output displacement and natural frequency, when $t_{b}$ is set to 1 mm and $l_{1}$ is 0.8 mm. This figure indicates that the output displacement decreases and natural frequency increases with the increase in $t_{c}$. However, as $l_{3}$ increase, the output displacement increases and natural frequency decreases, where the hinge of the lever mechanism is closer to the lower-bridge mechanism.

Figure 11 shows the effects of $t_{b}$ and $t_{c}$ on the output displacement and natural frequency, where $l_{1}$ is set to 0.8 mm and $l_{3}$ is set to 2 mm. Three conclusions can be drawn as follows:There is an optimal value of $t_{b}$ (1.1 mm), for a fixed value of $t_{c}$, to make the HBLB maximum output displacement. This is similar to the conclusions drawn from Figure 9.The output displacement of the HBLB increases with a decrease in $t_{c}$ when $t_{b}$ is fixed and it is greater than 0.8 mm.The natural frequency of the HBLB increases with an increase in either $t_{b}$ or $t_{c}$.

Based on the above conclusions, the lower-bridge and the lever mechanisms jointly form the first amplifier of the HBLB. The lower-bridge mechanism plays a significant role in the output displacement. Moreover, the HBLB had a larger output displacement when $t_{b}$ was equal to 1.1 mm. However, the output displacement of the HBLB decreased with the increase in the lever-mechanism hinge thickness when the hinge thickness of the lower-bridge mechanism was greater than 0.8 mm (according to the simulation). Moreover, the lever mechanism plays a vital role in the natural frequency. An increase in $t_{c}$ and a decrease in $l_{3}$ can both improve the natural frequency of the HBLB effectively.

The optimal geometric parameters of the HBLB mechanism are listed in Table 4, based on the above simulation analysis.

### 4.2. Mechanical Performance

The output stiffness and amplification ratio obtained by the analytical model were verified by FEA simulation. An input force range of 0–400 N was applied at the input to obtain the relevent displacement $x_{in}$ and $y_{out}$. The output stiffness of the theoretical model was 1.57 N/μm, whereas the simulation result was 1.61 N/μm; the error was less than 3%. The amplification ratio k, calculated using Equation (Equation 16), was 8.29, whereas the simulation value was 8.74, where the error was less than 6%. The error mainly comes from the simplified theoretical model ignoring the effect of shear force on the deformation of the mechanism. The developed mathematical model of the HBLB mechanism corresponded with the simulation results, as shown in Figure 12. Furthermore, the proposed amplifier mechanism showed an ideal amplification ratio, compared to other amplifier mechanisms with different structures, as presented in Table 5.

A force of 400 N was exerted to the input ends to determine the maximum stress of the HBLB mechanism. Figure 13 shows that the maximum stress appeared at H joint. The maximum stress, 173.8 MPa, is lower than the yield stress of 445 MPa for the chosen material, indicating that the HBLB mechanism will always operate in the elastic deformation range throughout the operating scope. The maximum output displacement was 239.7 μm.

Modal analyses of the HBLB mechanism was also performed using FEA. Owing to the case of PZT installation, the piezo-stack can be seen as a spring with a stiffness of 49 N/μm and a certain mass in the simulation model [31]. Figure 14 shows the first four modal vibration shapes. The first mode (operational modal) appeared at approximately 1.23 kHz. The natural frequency of the proposed amplifier mechanism is higher than those of other amplifier mechanisms with different structures, as presented in Table 5, which effectively supports the high-performance piezo jet dispenser.

## 5. Experimental Validation

Experiments are conducted to evaluate the performance of the piezoelectric actuator. The mechanism is made via wire electrical-discharge machining using Al7075. The overall size of the mechanism is 75 mm × 49 mm × 14 mm. A piezo-stack (120 V/7 × 7 × 38 mm, from COREMORROW) is assembled inside the mechanism, and a power amplifier with a high performance (the amplification ratio is 10, the bandwidth is 10 kHz, and the voltage output range is −30 V to 150 V), which is designed to drive the piezoelectric actuator. The signal is generated by Matlab/Simulink real-time control package xPC Target and NI I/O hardware (PCI-6259). Two laser sensors (Type LK-H020, Keyence) measure the input and output displacements through the sensor target. The displacement data are recorded using a LK-navigator controller powered by a DC power supply. The experimental setup is shown in Figure 15.

Triangle wave voltages with the maximum voltages of 30 V, 60 V, 90 V, and 120 V are applied into the piezoelectric actuator, and the experimental results are shown in Figure 16a. The test results show the hysteresis characteristics of the piezoelectric actuator under an open loop operation, which is significant for open-loop control. Figure 16b shows the input and output sinusoidal motions measured at a voltage of 120 V. The magnification ratio of the amplifier mechanism was 8.17, which is close to the theoretical (8.29) and simulated results (8.74).

As shown in Figure 16c, the stroke range test results show that the fabricated piezoelectric actuator has a maximum stroke range of 223.2 μm under a trapezoid voltage signal with a slope of 120 V/s and peak voltage of 120 V, which differs from the simulated value by 6.9%. The deviation range of experimental data from experimental result is 19 μm, as shown in Figure 17b, and the relative standard deviation is under 3.0%. This error could be caused by the preload method, where the piezoelectric stack is not a full stroke output because the thread is squeezed by a large input force.

To verify the dynamic performance of the piezoelectric actuator, a sinusoidal sweep excitation is used by independently applying a sine input voltage with a fixed amplitude of 400 mV and varying the frequency between 1–2000 Hz. The resonant frequency test result is shown in Figure 16d. The first natural frequency is approximately 1.184 kHz, which agrees well with the finite element simulation results. The deviation range of experimental data from experimental result is 141 Hz, as shown in Figure 17a, and the relative standard deviation is under 3.8%. The relative deviation between the test results and the finite element simulation results is 3.9%.

## 6. Conclusions

This study presents the design, test, and analysis of a hybrid displacement amplifier to support high-performance piezo-jet dispenser. The hybrid bridge-lever-bridge mechanism (HBLB) solved the issue of low frequency bandwidth on the traditional bridge-type amplifier mechanism. By combining the input end of BTAM with lever mechanism, the resonant frequency has been significantly increased without sacrificing the output displacement of the HBLB mechanism. Subsequently, an accurate analytical model that describes the force-displacement relationship considering the lateral deformation of the input end was established Finally, a machined prototype of the proposed HBLB mechanism was fabricated and evaluated its motion behaviors and dynamic performances. The experimental results showed that the working bandwidth was higher than existing designs and the motion range satisfied the jetting requirement,. In future research, a high-performance piezo jet dispenser based on the HBLB mechanism will be investigated, where the analysis and control of dispensing system will be more attractive.

## Figures and Tables

**Figure 1 micromachines-14-00322-f001:**
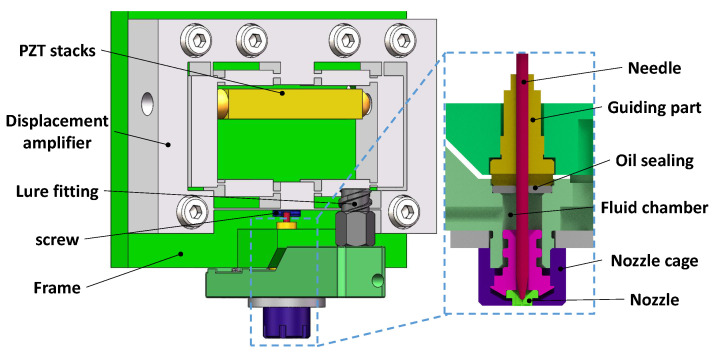
Configuration of a jet dispenser.

**Figure 2 micromachines-14-00322-f002:**
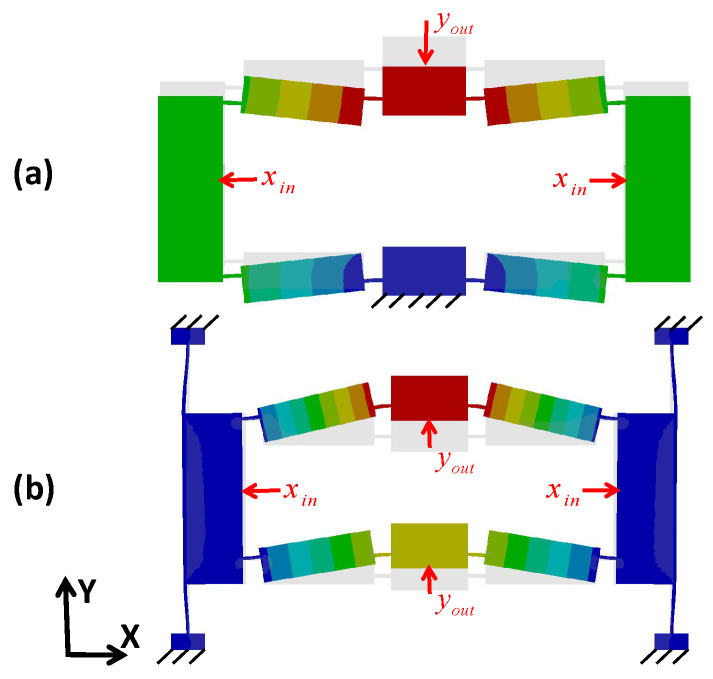
Schematic of (**a**) the BTAM, and (**b**) the improved bridge-type mechanism.

**Figure 3 micromachines-14-00322-f003:**
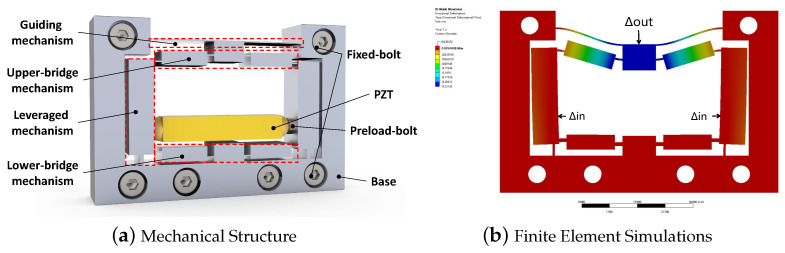
Schematic of the HBLB.

**Figure 4 micromachines-14-00322-f004:**
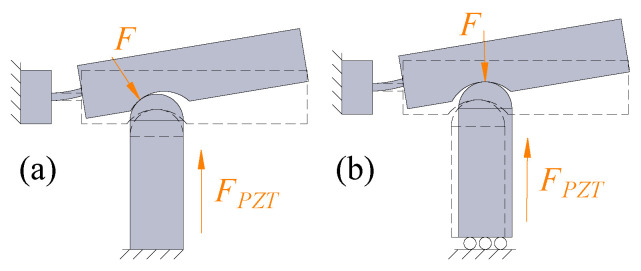
Comparison of different contact ways of a piezoelectric actuator: (**a**) traditional lever mechanism and (**b**) designed hybrid mechanism.

**Figure 5 micromachines-14-00322-f005:**
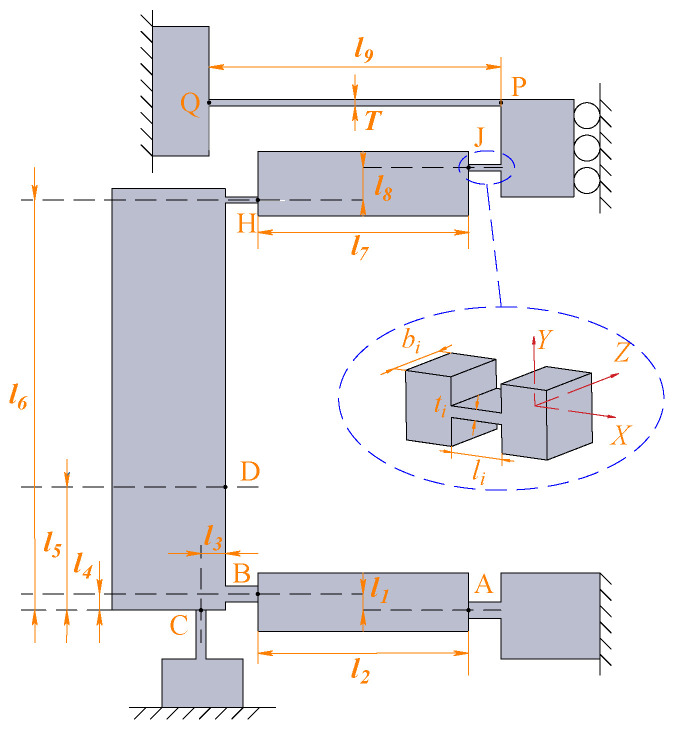
Structures of the half HBLB mechanism and flexure hinge.

**Figure 6 micromachines-14-00322-f006:**
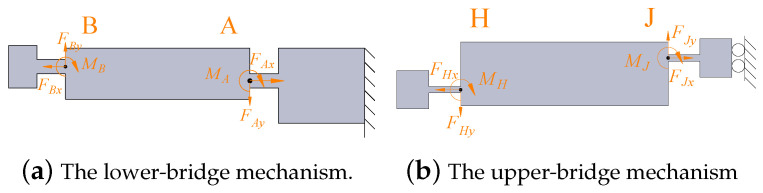
Static body analysis of the HBLB.

**Figure 7 micromachines-14-00322-f007:**
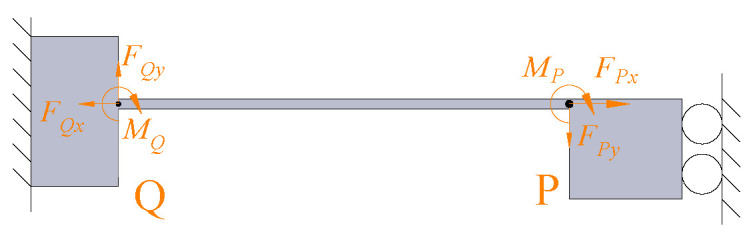
Static body analysis of the guiding beam.

**Figure 8 micromachines-14-00322-f008:**
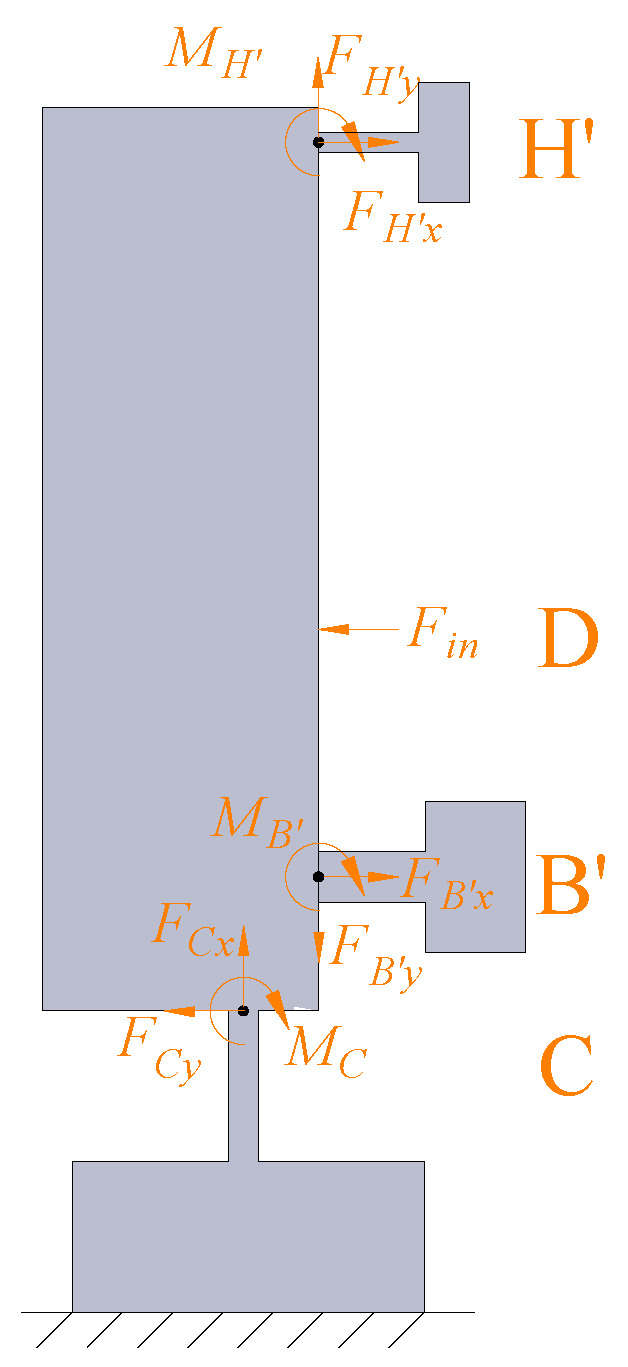
Static body analysis of the input end.

**Figure 9 micromachines-14-00322-f009:**
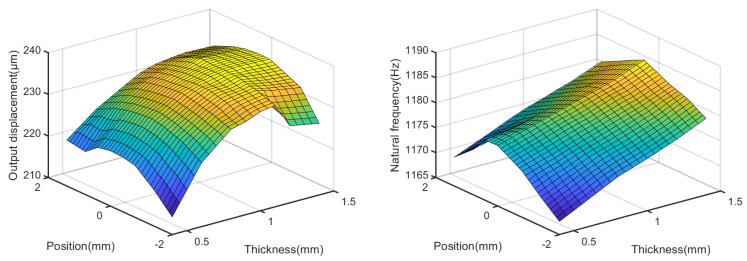
Effects of $t_{b}$ and $l_{1}$ on the output displacement and natural frequency.

**Figure 10 micromachines-14-00322-f010:**
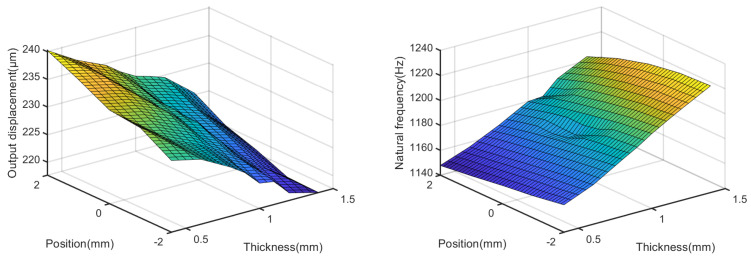
Effects of $t_{c}$ and $l_{3}$ on the output displacement and natural frequency.

**Figure 11 micromachines-14-00322-f011:**
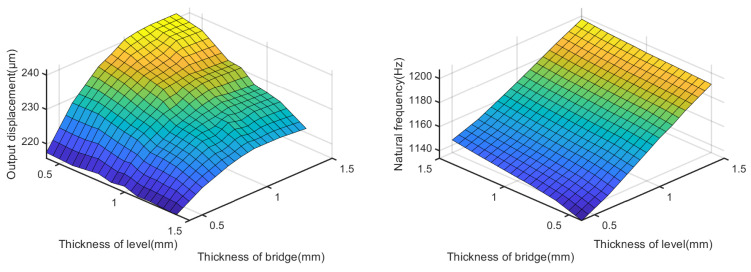
Effects of $t_{b}$ and $t_{c}$ on the output displacement and natural frequency.

**Figure 12 micromachines-14-00322-f012:**
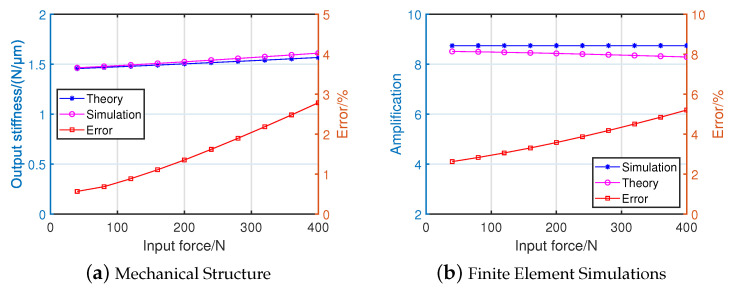
Schematic of the HBLB.

**Figure 13 micromachines-14-00322-f013:**
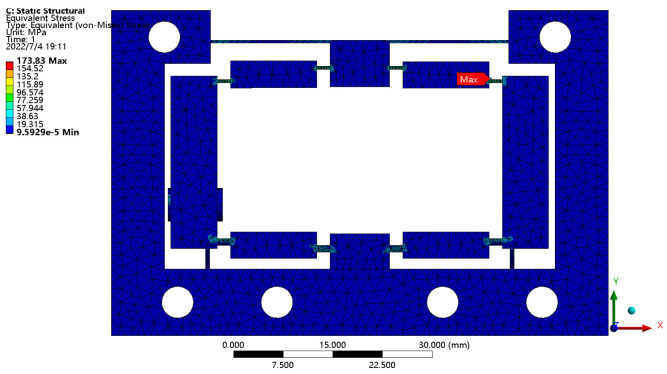
Stress distribution of the mechanism under the maximum input force.

**Figure 14 micromachines-14-00322-f014:**
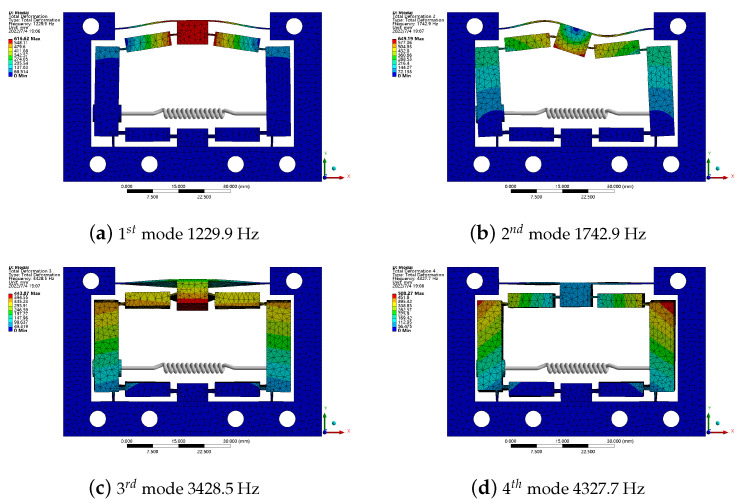
The first four order mode vibration shapes of the HBLB.

**Figure 15 micromachines-14-00322-f015:**
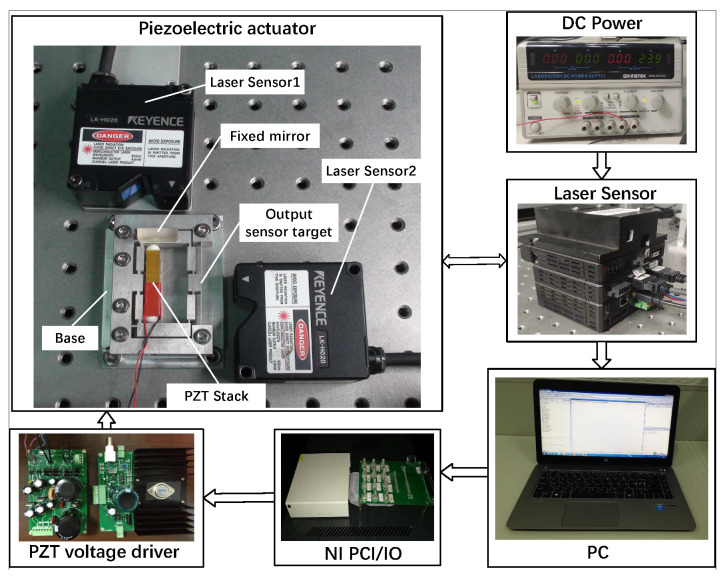
The schematic of the testing experimental system.

**Figure 16 micromachines-14-00322-f016:**
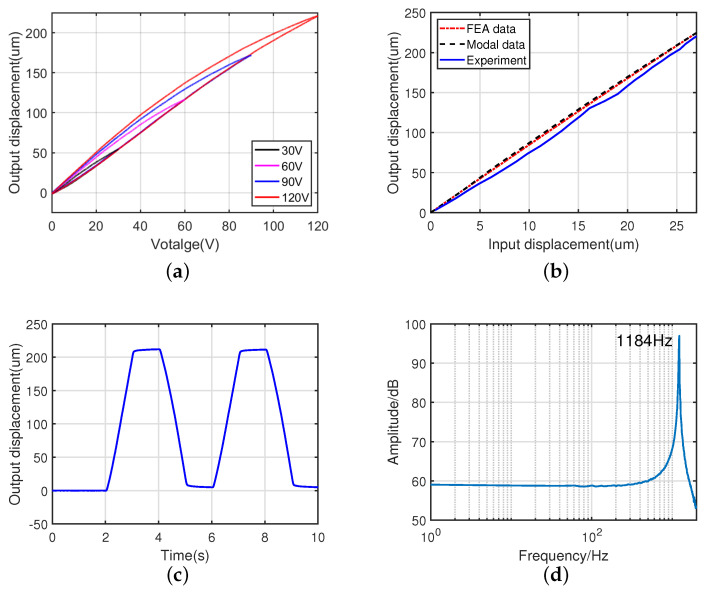
Experimental results of the machined prototype. (**a**) Hysteresis experiment. (**b**) Amplification ratio test. (**c**) Stroke range test. (**d**) Resonant frequency test.

**Figure 17 micromachines-14-00322-f017:**
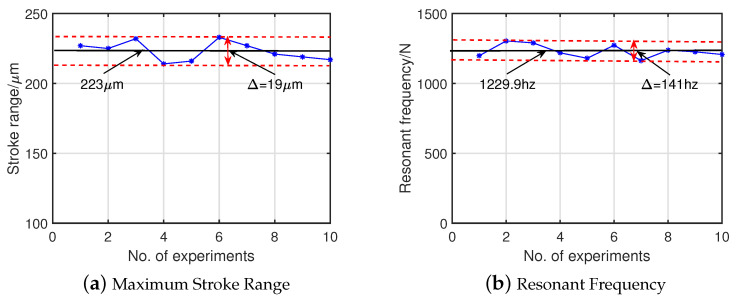
Ten sets of experimental data from test experiments.

**Table 1 micromachines-14-00322-t001:** Values of structural parameters.

$t_{b}$ (mm)	$l_{1}$ (mm)	$t_{c}$ (mm)	$l_{3}$ (mm)
0.4–1.4	−2–1.4	0.4–1.4	1.5–5.5

**Table 2 micromachines-14-00322-t002:** The material of the hybrid mechanism of Al7075.

Young’s Modulus	Yield Strength	Density	Poisson’s Ratio
71 GPa	455 MPa	2810 kg/m${}^{3}$	0.33

**Table 3 micromachines-14-00322-t003:** The technical parameters of adopted piezo-stack.

Dimension	Blocking Force	Nominal Displacement	Voltage	Stiffness	Frequency
7 × 7 × 36 mm	1960 N	38 μm	120 V	49 N/μm	35 kHz

**Table 4 micromachines-14-00322-t004:** Main parameters of the HBLB mechanism.

Para of structure	$l_{1}$	$l_{2}$	$l_{3}$	$l_{4}$	$l_{5}$	$l_{6}$	$l_{7}$	$l_{8}$	$l_{9}$
Value (mm)	1	13	1.5	1	6.7	25.3	13	2	18
Para of structure	$t_{b}$	$l_{b}$	$b_{b}$	$t_{c}$	$l_{c}$	$b_{c}$	$t_{h}$	$l_{h}$	T
Value (mm)	1	2	10	0.6	3	10	0.5	2	6.5

**Table 5 micromachines-14-00322-t005:** Critical parameters of different amplifier mechanisms.

Ref.	Type	Amplification Ratio	Natural Frequency
1 [19]	Scott-Russell with half-bridge(SRHB)	5.7	667 Hz
2 [31]	Scott-Russell compound bridge-type(SRCBT)	8.54	248 Hz
3 [6]	L-shape levers and half-bridge(LHLSB)	8.8	484 Hz
4 [30]	Arch-Shape Bridge Type	6.49	611.9 Hz
5 [27]	Diamond-type micro-displacement amplifier	7.8	342 Hz
6 [29]	Displacement amplifying mechanism	7.78	469 Hz
7 [37]	Lever amplification	6.75	886 Hz
	This work	8.74	1229.9 Hz

## Data Availability

The data presented in this study are available from the corresponding author upon request.

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
