# Peer review of "Design and Analysis of a Hybrid Displacement Amplifier Supporting a High-Performance Piezo Jet Dispenser"

_micromachines, 2023, doi:10.3390/mi14020322_

Round 1

Reviewer 1 Report

In this study, a HBLB compliant piezoelectric amplifier is designed and developed. The analytical model is developed and its verification through a finite element analysis (FEA) is carried out by the authors.

The paper is well written and establishes the novelty of proposed model in terms of high frequency operation up to 1.184kHz. However, following points must be considered before publication of the manuscript in the micromachines journal.

I suggest major revision of the manuscript based on following comments before its publication.

·        Please incorporate the experimental results for the jetting performance of the dispenser. The developmental phase takes long time, however, simple experimental validation of jetting performance would prove the novelty and claims of this research article.

Following article may shed some light on droplet formation mechanism:

 Journal of Nanoscience and Nanotechnology 19(3):1843-1847 DOI: 10.1166/jnn.2019.16223

·        Please include the details and specifications of designed power amplifier, as it plays a vital role during performance analysis of the jet dispensers.

·        Can the simulated results be reproduced for real prototype, if yes, then how can we exert a force of 400N on the prototype by piezoelectric actuator?

·        Please include the response characteristics of adopted piezo-stack in terms of performance, like blocking force, displacement, voltage, and frequency to validate the comparison with the simulated model.

·        Please briefly mention the reason why there was a difference of about 3% for output stiffness value and about 6% difference for the amplification ratio, as for solid mechanics, these differences should be justified.

·        A lot of work is already reported on this topic, so please enrich the manuscript with further relevant references and highlight the novelty of current work.

·        Please rewrite the legend of figure 15 correctly, by rephrasing it.

·        Please change the expression to "machined prototype" in figure 16’s legend.

Author Response

Response to Reviewer 1 Comments

Point 1: Please incorporate the experimental results for the jetting performance of the dispenser. The developmental phase takes long time, however, simple experimental validation of jetting performance would prove the novelty and claims of this research article.

Following article may shed some light on droplet formation mechanism:

 “Journal of Nanoscience and Nanotechnology 19(3):1843-1847 DOI: 10.1166/jnn.2019.16223”.

Response 1: Note that the main contribution of this work is the development of a new type displacement amplification mechanism with high natural frequency supporting jet dispensing applications, instead of a dispenser system. Therefore, the experiments were carried out to verify the mechanical performance of the proposed displacement amplifier. To this end, the workspace, amplification ratio and resonant frequency were particularly tested, which outperformed existing designs in recent literature and satisfied the requirements of jetting. The validation of jetting performance, as suggested by the reviewer, is certainly important but out of the scope of the present work. Note that the jetting performance is also dependent on other factors, such as the dispenser nozzle, actuator waveform, as well as the operation frequency. Therefore we are working on the integration of the present result to a optimized jet dispenser system to comprehensive improve the jetting performance, which will be reported separately.

Point 2: Please include the details and specifications of designed power amplifier, as it plays a vital role during performance analysis of the jet dispensers.

Response 2: We took this suggestion and provided more details of designed power amplifier: the amplification ratio is 10, the bandwidth is 10 kHz and the voltage output range is -30V to 150V. Please refer to the modifications in Section 5.

Point 3: Can the simulated results be reproduced for real prototype, if yes, then how can we exert a force of 400N on the prototype by piezoelectric actuator? 

Response 3: Thanks for the comment. The simulated results can be reproduced for real prototype. Based on the stiffness and electromechanical transforming coefficient of adopted piezo-stack and the input stiffness of the displacement amplifier mechanism, we can calculate the actual operating output force of the adopted piezo-stack in case of certain driving voltage. Correspondingly, we exerted a force of 400N on the prototype by piezoelectric actuator by applying a voltage of 120V to the piezo-stack.

Point 4: Please include the response characteristics of adopted piezo-stack in terms of performance, like blocking force, displacement, voltage, and frequency to validate the comparison with the simulated model.

Response 4: We took the reviewer’s suggestion and added the technical parameters of adopted piezo-stack, including the dimension size, blocking force, nominal displacement, maximum voltage, stiffness and frequency. Please refer to Table 3.

Point 5: Please briefly mention the reason why there was a difference of about 3% for output stiffness value and about 6% difference for the amplification ratio, as for solid mechanics, these differences should be justified.

Response 5: Thanks for the comment. The error mainly comes from the simplified theoretical model ignoring the effect of shear force on the deformation of the mechanism. In this revision, we added the error analysis on this problem. Please refer to Section 4.2.

Point 6: A lot of work is already reported on this topic, so please enrich the manuscript with further relevant references and highlight the novelty of current work.

Response 6: Thanks for the comment. We added some relevant references on this topic and we have rephrased the introduction to highlight the novelty of current work, please refer to introduction section.

Point 7: Please rewrite the legend of figure 15 correctly, by rephrasing it.

Response 7: We took the suggestion and modified Figure 15.

Point 8: Please change the expression to "machined prototype" in figure 16’s legend.

Response 8: Done.

Reviewer 2 Report

Although the authors have presented an interesting research and the research area is also significant, certain improvements should be incorporated in the manuscript before it can be published as a journal article:

1. The authors should briefly describe the working of the design presented by correlating it with the physics behind the piezo-electric effect. How do individual elements react to electrical flow and how does the mechanism transduce electrical energy to mechanical energy.

2. The authors have extensively used abbreviations (e.g. PZT). A list of nomenclature should be included before the introduction section. 

3. In results section the authors should outline the repeatability of their results.

4. The authors should present a detailed design on how the mechanism will be incorporated in the jet dispenser.

5. The authors should provide quantitative data to highlight the advantages of the proposed HBLB mechanism over existing designs.

6. The conclusions section should be revisited and revised to highlight the novelty of this research and its potential benefits.

Author Response

Response to Reviewer 2 Comments

Point 1: The authors should briefly describe the working of the design presented by correlating it with the physics behind the piezo-electric effect. How do individual elements react to electrical flow and how does the mechanism transduce electrical energy to mechanical energy. 

Response 1: We appreciate the comments. The description on the working of the design presented is an important part of the introduction section. We added information about piezo-electric effect and the energy transfer process of the designed mechanism. Please refer to introduction section.

Point 2: The authors have extensively used abbreviations (e.g. PZT). A list of nomenclature should be included before the introduction section.

Response 2: We appreciate the comment. PZT is an abbreviation for Piezoactuator, and we listed the abbreviations and their nomenclature used in the article following the journal paper format. Please refer to Abbreviations.

Point 3: In results section the authors should outline the repeatability of their results.

Response 3: We appreciate this valuable comment and agree with the reviewer that the repeatability is essential for experimental results. As a matter of fact, each experiment was conducted ten times with consistent operating conditions to guarantee the reliability of experimental results, which demonstrated the relative standard deviation less than 3.0% and 3.8%. We outline the repeatability of the testing results and added a figure to show the repeatability in Section 5.

Point 4: The authors should present a detailed design on how the mechanism will be incorporated in the jet dispenser.

Response 4: We took the reviewer’s suggestion and added some design details on how the amplification mechanism incorporated in the jet dispenser. A new figure was also replaced in Fig. 1 to illustrated the working principle. Please refer to Section 2.1. 

Point 5: The authors should provide quantitative data to highlight the advantages of the proposed HBLB mechanism over existing designs.

Response 5: We appreciate the reviewer’s suggestion and provided quantitative comparisons between the proposed HBLB mechanism with existing designs in recent literature. Please refer to Table. 5. 

Point 6: The conclusions section should be revisited and revised to highlight the novelty of this research and its potential benefits.

Response 6: Thanks for this good suggestion. In this revision, we significantly revised the conclusion to further highlight the novelty and the potential benefits of this research. Please refer to conclusions section.

Reviewer 3 Report

In this manuscript, the author proposed a displacement amplifier mechanism, a hybrid bridge-lever-bridge (HBLB), to address the high-performance dispensing operation issue. The output stiffness as well as a guaranteed large amplification ratio was enhanced by adding a guiding beam. The finite element analysis (FEA) was used to describe the full elastic deformation behavior of the HBLB mechanism and optimize the structural parameters. I suggest the publication of this manuscript in Micromachines, a high-performance journal if the author can address the following concerns appropriately.

1. The author should add scale bars to the finite element simulation results of Figure 3b.

2. In the introduction section of this manuscript, the author should concentrate more on the novelty of this project, and the logic should be well organized.

3. The author prepared a high-performance piezo jet dispenser with a displacement amplifier mechanism. Likewise, materials with piezoresistive behavior are also of significance to boost the development of flexible electronics (DOI: 10.1016/j.apsusc.2022.153803; DOI: 10.1088/1361-6463/aa84a3; DOI: 10.1021/acsami.2c14907; DOI: 10.1063/5.0083278). The author can compare the mechanism, feature, and application of piezoelectric and piezoresistive materials.

Author Response

Response to Reviewer 3 Comments

Point 1: The author should add scale bars to the finite element simulation results of Figure

3b.  

Response 1: We took the reviewer’s suggestion and added scales bars to the finite element simulation results. Please refer to Figure 3b in this revision.

Point 2: In the introduction section of this manuscript, the author should concentrate more on the novelty of this project, and the logic should be well organized.

Response 2: Thanks for this comment. In this revision, we reorganized the description of the introduction and emphasized the novelty of this work. Please refer to introduction section.

Point 3: The author prepared a high-performance piezo jet dispenser with a displacement amplifier mechanism. Likewise, materials with piezoresistive behavior are also of significance to boost the development of flexible electronics (DOI: 10.1016/j.apsusc.2022.153803; DOI: 10.1088/1361-6463/aa84a3; DOI: 10.1021/acsami.2c14907; DOI: 10.1063/5.0083278). The author can compare the mechanism, feature, and application of piezoelectric and piezoresistive materials.

Response 3: We appreciate the comment and agree with the reviewer that the Piezoelectric materials play an important role in flexible electronics, precision engineering and microelectronics packaging. We added some comparisons about the application of piezoelectric and piezoresistive materials. Based on this, we have rephrased the introduction, thank you again for your valuable suggestions.

Round 2

Reviewer 2 Report

All reviewer suggestions have successfully been addressed. The article may be published in this esteemed journal.